# A Policy-Guided Imitation Approach for Offline Reinforcement Learning

**Haoran Xu**♠* **Li Jiang**♣* **Jianxiong Li**♣ **Xianyuan Zhan**♣,◇
♠JD Technology, Beijing, China
♣Tsinghua University, Beijing, China
◇Shanghai AI Laboratory, Shanghai, China
{ryanxhr,jiangli3859}@gmail.com

## Abstract

Offline reinforcement learning (RL) methods can generally be categorized into two types: RL-based and imitation-based. RL-based methods could in principle enjoy out-of-distribution generalization but suffer from erroneous off-policy evaluation. Imitation-based methods avoid off-policy evaluation but are too conservative to surpass the dataset. In this study, we propose an alternative approach, inheriting the training stability of imitation-style methods while still allowing logical out-of-distribution generalization. We decompose the conventional reward-maximizing policy in offline RL into a guide-policy and an execute-policy. During training, the guide-policy and execute-policy are learned using only data from the dataset, in a supervised and decoupled manner. During evaluation, the guide-policy guides the execute-policy by telling where it should go so that the reward can be maximized. By doing so, our algorithm allows *state-compositionality* from the dataset, rather than *action-compositionality* conducted in prior imitation-style methods. We dumb this new approach Policy-guided Offline RL (`POR`). `POR` demonstrates the state-of-the-art performance on D4RL, a standard benchmark for offline RL. We also highlight the benefits of `POR` in terms of improving with supplementary suboptimal data and easily adapting to new tasks by only changing the guide-policy. Code is available at https://github.com/ryanxhr/POR.

## 1 Introduction

Offline RL, also known as batch RL, allows learning policies from previously collected data, without online interactions [29, 31]. It is a promising area for bringing RL into real-world domains, such as robotics [21], healthcare [47] and industrial control [62]. In such scenarios, arbitrary exploration with untrained policies is costly or dangerous, but sufficient prior data is available. While most off-policy RL algorithms are applicable in the offline setting by filling the replay buffer with offline data, improving the policy beyond the level of the behavior policy often entails querying the value function (i.e., Q function) about values of actions that were not seen in the dataset. Those out-of-distribution actions can be deemed as adversarial examples of the Q function [28], which cause extrapolation error of the Q-function. The error will be accumulated by the deadly triad issue [50], propagate across the state-action space through the iterative dynamic programming procedure [4], and can not be eliminated without requiring a growing batch of online samples [31].

To alleviate this issue, prior model-free offline RL methods typically add a behavior regularization term to the policy improvement step, to limit how far it deviates from the behavior policy. This can be achieved explicitly by calculating some divergence metrics [26, 54, 38, 13], or implicitly

---

*Equal contribution. Correspondence to Haoran Xu, Xianyuan Zhan.

36th Conference on Neural Information Processing Systems (NeurIPS 2022).

by regularizing the learned value functions to assign low values to out-of-distribution actions [28, 24, 1, 2]. Nevertheless, this imposes a trade-off between accurate value estimation (more behavior regularization) and maximum policy improvement (less behavior regularization). To avoid this problem, recently a branch of methods bypass querying the values of unseen actions by performing some kind of imitation learning on the dataset[*]. This can be achieved by filtering trajectories based on their return [40, 8], or reweighting transitions based on how advantageous they could be under the behavior policy [4, 25], or just be conditioned on some variables without any dataset reweighting [7, 10]. Although imitation-style methods enjoy a stable training process and are able to effectively perform multi-step dynamic programming by assigning the proper reweighting weight or conditioned variable [25, 10], they only allow *action-compositionality* from the dataset, lose the ability to surpass the dataset by out-of-distribution generalization, which only appears in RL-based methods.

In this work, we propose an alternative approach. We aim at inheriting the training stability of imitation-style methods while still allowing logical out-of-distribution generalization. To do so, we propose Policy-guided Offline RL (POR), an algorithm that employs two policies to solve tasks: a guide-policy and an execute-policy. The job of the guide-policy is to learn the optimal next state given the current state, and the job of the execute-policy is to learn how different actions can produce different next states, given the current state. By this manner, we decompose the original task of RL (i.e., reward-maximizing) into two distinct yet complementary tasks, and this decomposition makes each task much easier to be solved. During evaluation, the guide-policy serves as a guide for the execute-policy by telling it where to go so that the reward can be maximized. By doing so, our algorithm allows *state-compositionality* from the dataset, rather than *action-compositionality* conducted in prior work, which owns logical out-of-distribution generalization.

Our method is easy to implement by only adding the learning of a guide-policy, the training stage of the guide-policy and execute-policy are decoupled, using in-sample learning from the dataset. We test POR in widely-used D4RL offline RL benchmarks and demonstrates the state-of-the-art performance, especially on low-quality datasets that require out-of-distribution generalization to achieve a high score. We also show that by decoupling the learning of guide-policy and execute-policy, we can enhance the guide-policy with supplementary suboptimal data, or re-learn the guide-policy to adapt to new tasks with different reward functions, without changing the execute-policy.

## 2 Related Work

**RL-based offline approach**     A large portion of offline RL methods is RL-based. RL-based methods typically augment existing off-policy methods (e.g., Q-learning or actor-critic) with a behavior regularization term. The primary ingredient of this class of methods is to propose various policy regularizers to ensure that the learned policy does not stray too far from the behavior policy. These regularizers can appear explicitly as divergence penalties [54, 26, 13], implicitly through weighted behavior cloning [52, 40, 38], or more directly through careful parameterization of the policy [14, 65]. Another way to apply behavior regularizers is via modification of the critic learning objective to incorporate some form of regularization to encourage staying near the behavioral distribution and being pessimistic about unknown state-action pairs [37, 28, 24, 58]. There are also several works incorporating behavior regularization through the use of uncertainty. The uncertainty quantification can be done via Monte Carlo dropout [55] or explicit ensemble models [2, 61, 22, 63]. Note that the behavior regularization weight is crucial in RL-based methods. With a small weight, RL-based methods could in principle enjoy out-of-distribution generalization since they perform true dynamic programming. However, a small weight will make erroneous off-policy evaluation due to distribution shift and the policy is extremely vulnerable to the value function misestimation.

**Imitation-based offline approach**     Another line of methods, on the contrary, performs some kind of imitation learning on the dataset, without the need to do off-policy evaluation. When the dataset is good enough or contains high-performing trajectories, we can simply clone the actions observed in the dataset [41], or perform some kind of filtering or conditioning to extract useful transitions. For instance, recent work filters trajectories based on their return [8, 40], or directly filters individual transitions based on how advantageous these could be under the behavior policy and then clones them [4, 25, 5, 16, 56]. Conditioned BC methods are based on the idea that every transition or trajectory

---

[*]The core difference between RL-based and imitation-based methods is that RL-based methods learn a value function of policy $\pi$ while imitation-based methods don't.

is optimal when conditioned on the right variable [15, 10]. In this way, after conditioning, the data becomes optimal given the value of the conditioned variable. The conditioned variable could be the cumulative return [7, 19, 27], or the goal information if provided [34, 15, 10, 59]. A recent study [10] shows that goal-conditioning can be super useful in D4RL AntMaze datasets. However, this introduces the assumption that we have some prior information about the structure of the task, which is beyond the standard offline RL setup.

Our method can be viewed as a combination of RL-based and imitation-based methods. We adopt an imitation-style manner to train the goal-policy and execute-policy. During evaluation, however, following the guide-policy, the execute-policy could produce out-of-distribution actions by leveraging the generalization capacity of the function approximator.

## 3    Preliminaries

**Offline RL**    We consider the standard fully observed Markov Decision Process (MDP) setting [46]. An MDP can be represented as $\mathcal{M} = (\mathcal{S}, \mathcal{A}, T, r, \rho, \gamma)$ where $\mathcal{S}$ is the state space, $\mathcal{A}$ is the action space, $T(\cdot|s, a)$ is the transition probability distribution function, $r(s, a)$ is the reward function, $\rho$ is the initial state distribution and $\gamma$ is the discount factor, we assume $\gamma \in (0, 1)$ in this work. The goal of RL is to find a policy $\pi(a|s) : \mathcal{S} \times \mathcal{A} \rightarrow [0, 1]$ that maximizes the expected cumulative discounted reward (or called return) along a trajectory as

$$\max_{\pi} \mathbb{E} \left[ \sum_{t=0}^{\infty} \gamma^t r(s_t, a_t) \middle| s_0 = s, a_0 = a, s_t \sim T(\cdot|s_{t-1}, a_{t-1}), a_t \sim \pi(\cdot|s_t) \text{ for } t \geq 1 \right]. \quad (1)$$

In this work, we focus on the offline setting. The goal is to learn a policy from a fixed dataset $\mathcal{D} = \left\{ \tau^i = \left( (s_0^i, a_0^i, s_0'^i, r_0^i), (s_1^i, a_1^i, s_1'^i, r_1^i), \cdots, (s_H^i, a_H^i, s_H'^i, r_H^i) \right) \right\}_{i=1}^N$ consisting of trajectories that are collected by different policies, where $H$ is the time horizon. The dataset can be heterogenous and suboptimal, we denote the underlying behavior policy of $\mathcal{D}$ as $\mu$, which represents the conditional distribution $p(a|s)$ observed in the dataset.

**RL via Supervised Learning**    Conventional RL methods generally either compute the derivative of (1) with respect to the policy directly via policy gradient methods [43, 20], or estimate a value function or Q-function by means of TD learning [53], or both [44, 17]. In the *RL via Supervised Learning* framework, we avoid the complex and potentially high-variance policy gradient estimators, as well as the complexity of temporal difference learning. Instead, we perform conditioned behavior cloning on some extra information, such as goal state, cumulative return, or language description [34, 15]. When applying the framework to the offline RL setting, we can take the offline dataset $\mathcal{D}$ as input and find the outcome-conditioned policy $\pi$ that optimizes

$$\max_{\pi} \mathbb{E}_{\tau \sim \mathcal{D}, t \sim \text{Unif}(1, H), \omega \sim g(\cdot|\tau_{t:H})} \left[ \log \pi(a_t|s_t, \omega) \right]. \quad (2)$$

where $\omega$ is an outcome conditioned on the remaining trajectory, i.e., $\omega \sim g(\cdot|\tau_{t:H})$, and we use $\tau_{i:j} = (s_i, a_i, \ldots, s_j)$ to denote a fragment of the trajectory.

## 4    Policy-guided Offline Reinforcement Learning

We now continue to describe our proposed approach, POR. To have a clear understanding of the out-of-distribution generalization of POR, we begin with a motivating example which demonstrates that by allowing state-compositionality on the dataset, the agent is able to generalize to a much better policy than doing action-compositionality. Then we describe how we accomplish this idea in the more general case and introduce the algorithm details of POR. Finally, we give a theoretical analysis of POR, we are interested to see how close the policy induced by POR and the optimal policy could be.

### 4.1    A Motivating Example

We first use a simple, didactic toy example to illustrate the importance of state-stitching for offline RL. Let's consider the navigation task shown in Figure 1, where the goal is to find the shortest path from the starting location (bottom-left corner) to the goal location (top-right corner) in the grid world. The state and action space in this task are discrete, the state consists of the $(x, y)$ coordinate

and the agent could choose to move to any of its nearby 8 grids (i.e., the action space is 8). The agent receives a reward of $0$ for entering the goal state and $-1$ for all other transitions. The offline dataset (green-arrowed lines) consists of two suboptimal trajectories from the start location to the goal location, plus transitions of taking random actions at random states.

We use the grid color (more saturated to purple means higher value) to show the value function $V(s)$ computed from the offline dataset. We compare two methods that use the value function differently. At a state $s_b \sim \mathcal{D}$, the first method chooses the action $a_b$ in the dataset that leads to $s_b'$ with the highest $V(s_b')$; the second method chooses the action $a$ that leads to $s'$ in the dataset with the highest $V(s')$, note that $a$ is not necessarily to be within the dataset. The first method is doing action "stitching" in the dataset while the second method is doing state "stitching". Although both methods are able to "stitch" suboptimal trajectories together from the dataset and successfully travel from the start location to the goal location, the action-stitching method takes more steps (11 steps) compared with the state-stitching method (7 or 8 steps), as shown in Figure 1.

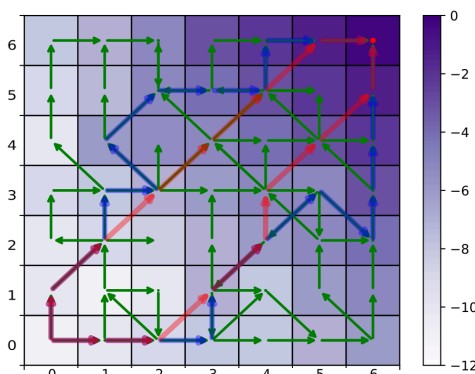

Figure 1: A toy example in the grid world that requires navigation from the start location (bottom-left corner) to the goal location (top-right corner). The offline dataset is colored with green, trajectories obtained from action-stitching method and state-stitching method are colored with blue and red, respectively. Note that both action-stitching and state-stitching methods have two equal-valued action choices at the start location ($\uparrow$ and $\rightarrow$), resulting in two different trajectories in both blue and red lines.

The action-stitching method introduced above is actually an abstract methodology of imitation-based methods, which imitate the transitons in the dataset unequally with different weight choices. However, these methods are too conservative to find the optimal trajectory, especially when there does not exist one in the offline dataset. To generate better-than-dataset trajectories (e.g., red-arrowed lines), the agent needs to take logical out-of-distribution actions.

Note that in the toy example, we give the agent the freedom to choose **any** action that leads to the highest-value state in the dataset. This allows the most out-of-distribution generalization and is realizable in some simple tasks (e.g., discrete state and action space) with high action coverage, for instance, in the toy example, given historical $(s, a, s')$ results at $(5, 4)$, the agent can successfully recognize that taking upper-right will arrive at $(6, 5)$. However, in the more general case, especially the continuous state and action space setting, this may make the agent do erroneously-generalized actions as the same state can hardly be observed twice. To perform logical out-of-distribution generalization, we need a *Prophet*, i.e., the guide-policy, to help the agent by telling it which state it should (high reward) and can (logical generalization) go to. It turns out that the involvement of the guide-policy also brings some theoretical meaning, we will discuss that in Section 4.4.

## 4.2 Learning the Guide-Policy

Recall that the goal of the guide-policy is to guide the execute-policy about which state $s$ it should go to. To accomplish that, we train a state value function $V(s) : \mathcal{S} \to \mathbb{R}$. The training of $V$ uses only $(s, s')$ samples in the offline dataset, it doesn't suffer from overestimation because there're no out-of-distribution actions involved. To approximate the optimal value function in the dataset, we adopt tricks from recent work [25, 35] by giving the $\ell_2$ loss with a different weight using expectile regression, yielding the following asymmetric $\ell_2$ loss

$$\min_{\phi} \ \mathbb{E}_{(s,r,s') \sim \mathcal{D}} \left[ L_2^{\tau} \left( r + \gamma V_{\phi'}(s') - V_{\phi}(s) \right) \right], \ \text{where} \ L_2^{\tau}(u) = |\tau - \mathbb{1}(u < 0)|u^2. \quad (3)$$

It can be seen that when $\tau = 1/2$, this operator is reduced to Bellman expectation operator, while when $\tau \to 1$, this operator approaches Bellman optimality operator. Note that our learning objective bears similarity with IQL [25], however, here we aim to learn a state value function while IQL aims to learn a state-action value function so as to extract the policy. To do this, IQL needs to learn both $Q$ and $V$ ($V$ is used to isolate the effect of state and approximate the expectile of $Q$ only with respect to the action distribution), the training of $V$ and $Q$ are coupled and may affect each other.

Simply maximizing the guide-policy with respect to $V(s)$ will result in a state where the execute-policy may make erroneous generalization. To alleviate this issue, we add a behavior cloning term to the learning objective of the guide-policy. Denote the guide-policy as $g_\omega(s) : \mathcal{S} \to \mathcal{S}$, the learning objective of $g$ is given by

$$\max_\omega \mathbb{E}_{(s,s')\sim\mathcal{D}} \left[ V_\phi(g_\omega(s)) + \alpha \log g_\omega(s'|s) \right], \tag{4}$$

where the weight $\alpha$ serves as the trade-off between guiding to space with high-reward and space that the execute-policy ought to have a correct generalization.

We also give an alternative learning objective that eliminates the need of $\alpha$ by using the residual $(r + \gamma V_{\phi'}(s') - V_\phi(s))$ as the behavior cloning weight, which we find work well in practice:

$$\max_\omega \mathbb{E}_{(s,s')\sim\mathcal{D}} \left[ e^{(r+\gamma V_{\phi'}(s')-V_\phi(s))} \log g_\omega(s'|s) \right]. \tag{5}$$

### 4.3 Learning the Execute-Policy: Training and Evaluation

After learning the guide-policy, now we turn to the execute-policy. Since the job of the execute-policy is to have a strong generalization ability, we adopt the *RL via Supervised Learning* framework by conditioning the execute-policy on $s'$ that encountered in the dataset. To be more specific, denote the execute-policy as $\pi_\theta(s, s') : \mathcal{S} \times \mathcal{S} \to [0, 1]$. During training, $\pi$ performs supervised learning by maximizing the likelihood of the actions given the states and next states, yielding the following objective

$$\max_\theta \mathbb{E}_{(s,a,s')\in\mathcal{D}} \left[ \log \pi_\theta(a|s, s') \right], \tag{6}$$

in some scenarios with low data quality, we also add the exponential residual weight to (6) to mitigate the effect of those bad actions (same as (5)).

During evaluation, given a state $s$, the final action is determined by both the guide-policy and the execute-policy, by

$$a = \arg\max_a \ \pi_\theta(a|s, g_\omega(s)). \tag{7}$$

Note that our learning objective (6) differs from previous *RL via Supervised Learning* methods in that we

---

**Algorithm 1** Policy Guided Offline RL

**Require:** $\mathcal{D}, \tau, \alpha$ (optional).
1: // Training
2: Initialize $V_\phi, V_{\phi'}, g_\omega, \pi_\theta$
3: **for** $t = 1, 2, \cdots, N$ **do**
4:     Sample transitions $(s, r, s') \sim \mathcal{D}$
5:     Update $V_\phi$ by Eq.(3)
6:     Update $g_\omega$ by Eq.(4) or Eq.(5)
7:     Update $V_{\phi'}$ by $\phi' \leftarrow \lambda\phi + (1 - \lambda)\phi'$
8: **end for**
9: **for** $t = 1, 2, \cdots, M$ **do**
10:     Sample transitions $(s, a, s') \sim \mathcal{D}$
11:     Update $\pi_\theta$ by Eq.(6)
12: **end for**
13: // Evaluation
14: Get initial state $s$, set $d$ as False
15: **while** not $d$ **do**
16:     Get action $a$ form Eq.(7)
17:     Roll out $a$ and get $(s', r, d)$
18:     Set $s = s'$
19: **end while**

---

only use the next state $s'$ as the conditioned variable, there's no need to estimate the cumulative return [7], which may be highly suboptimal in certain cases [11, 3]. Our method also works in settings where we don't know the goal information [10]. Also, during evaluation, previous methods show that the choice of conditioned variable is crucial important as little changes will cause significant performance difference [7]. In our method, owing to the existence of the guide-policy, the optimal conditioned variable can be automatically generated.

Our final algorithm, POR, consists of three supervised stages: learning $V$, learning $g$, and learning $\pi$. We summarize the training and evaluation procedure of POR in Algorithm 1. Note that the training of $g$ and $\pi$ are fully decoupled, this brings POR some nice properties to improve with supplementary data or transfer to new tasks, we will further investigate it in the experiments.

### 4.4 Analysis

In this section, we give a theoretical analysis of POR. Concretely, we aim to 1) give the lower bound of the performance difference between POR and the optimal policy $\pi^*$, and 2) analyze how the guide-policy $g$ will influence this bound.

We begin by introducing the notation and assumptions used in our analysis. We denote $P(s, s') : \mathcal{S} \times \mathcal{S} \to \mathcal{A}$ as the *inverse transition operator*. We denote $a_g$ and $a$ as the ground truth of $\pi(s, g(s))$ and $\pi(s, s')$, respectively. We also denote $\epsilon := \sup_{(s,a,s')\in\mathcal{D}} \|\pi(s, s') - a\|$ as the upper bound of the approximation error in the dataset during training. We then introduce the following two assumptions.

**Assumption 1.** *(Non-lazy MDP) The inverse transition operator $P(s, s')$ is $L_1$-Lipschitz continuous, i.e., $\|P(s_1, s_1') - P(s_2, s_2')\| \leq L_1 \|(s_1, s_1') - (s_2, s_2')\|$.*

**Assumption 2.** *(Lipschitz continuous function approximators) The execute-policy we trained is $L_2$-Lipschitz continuous, i.e., $\|\pi(s_1, s_1') - \pi(s_2, s_2')\| \leq L_2 \|(s_1, s_1') - (s_2, s_2')\|$.*

Assumption 1 holds when actions do have effects on state transitions (i.e. if $a \neq 0$, then $\|s' - s\| \geq \epsilon_s > 0$). We refer the MDP that satisfies Assumption 1 as *non-lazy MDP* and others as *lazy MDP*. It should be mentioned that most real-world tasks belong to the *non-lazy MDP* case, it is meaningless to study under the *lazy MDP* case because in *lazy MDP*, action changes will have little effect on state transitions, making the policy learning in vain. Assumption 2 is a mild assumption that is frequently utilized in plenty of works [51, 36, 32].

Then in Theorem 1, we show that under these two assumptions, the gap between the optimal action and the action output by POR can be bounded.

**Theorem 1.** *(Single step gap to optimal action). The single-step gap between optimal action and the action induced by our method can be bounded as*

$$\|\pi(s, g(s)) - a^*\| \leq \underbrace{(L_1 + L_2)\|g(s) - s'\|}_{l_1} + \underbrace{\|a_g - a^*\|}_{l_2} + \underbrace{\epsilon}_{l_3} \tag{8}$$

Theorem 1 states that single step optimal gap is related to three parts: $l_1, l_2, l_3$. $l_1$ is the generalization performance factor. It can be found that a small value of $\|g(s) - s'\|$ enables a small $l_1$, which indicates the necessity to constrain the guide-policy to stay close to the dataset. $l_2$ denotes the suboptimality constant. Intuitively, a loosened constraint on the guide-policy may induce a small suboptimality constant but on the other hand, may suffer from the risk of a high value of $l_1$. The guide-policy $g$ serves as a trade-off between $l_1$ and $l_2$, we could balance the constraint strength and suboptimality by adjusting $\alpha$ in Eq.(4) to obtain the lowest bound. $l_3$ is related to the approximation error on the training data, which can be reduced via improving the representation ability of function approximators but may suffer from a high $L_2$ due to the overfitting caused by over-parameterization. $l_1$ also depends on the Lipschitz constant $L_1$ and $L_2$. Note that a smooth *inverse transition operator* will result in a small value of $L_1$ and $L_2$ (because $\pi(s, s')$ is trained to approximate $P(s, s')$), this means our method may achieve better performance under a smoother inverse transition.

Based on Theorem 1, we can also give the performance bound of our method by replacing the $\epsilon$ in Theorem 3.2 in [15] with the RHS of Eq.(8): $J(\pi^*) - J(\pi) \leq \sup_{s,a,h}(l_1 + l_2 + l_3)T$, where $T$ is the horion of each episode, $h$ is the time step.

## 5 Experiments

We present empirical evaluations of POR in this section. We first evaluate POR against other baseline algorithms on D4RL [12] benchmark datasets. We then explore deeper on the guide-policy about the benefits of the decoupled training process. We finally establish ablation studies on the execute-policy.

### 5.1 Comparative Experiments on D4RL Benchmark Datasets

We first evaluate our approach on D4RL MuJoCo and AntMaze datasets [12]. Notice that most of MuJoCo datasets (except `random` and `medium` datasets) consist of a large fraction of near-optimal trajectories, while AntMaze datasets contain few or no near-optimal trajectories. Hence those low-quality datasets can serve as a good testbed for quantifying the *compositionality* ability.

We compare POR with both RL-based and imitation-based baselines. For RL-based baselines, we select the best one from 5 state-of-the-art algorithms, including BCQ [14], BEAR [26], BRAC [54], CQL [28] and TD3+BC [13]. For imitation-based baselines, we further split them into *Weighted BC* methods and *Conditioned BC* methods. Weighted BC methods include 10%BC [7], One-step RL [4] and IQL [24]. 10%BC is a filtered version of BC that runs behavior cloning on only the top 10% high-return trajectories in the dataset. One-step RL and IQL can be viewed as using different weights to do behavior cloning (see Appendix A for discussion). Conditioned BC methods include DT [7] and RvS [10]. DT is built with Transformer and conditioned on the cumulative return. RvS well studied the effect of different conditional information on different tasks. The score of RvS is

Table 1: Averaged normalized scores of POR against other baselines. The scores are taken over the final 10 evaluations with 5 seeds. In the *Best RL Baseline* column, the algorithm with the best performance among 5 RL-based algorithms (BCQ [14], BEAR [26], BRAC [54], CQL [28], TD3+BC [13]) is presented, where the previous four are took from their original papers and the last one is from our implementation. POR achieved the highest scores in 11 out of 18 tasks.

| D4RL Dataset | Weighted BC | | | Conditioned BC | | | Best RL Baseline |
|---|---|---|---|---|---|---|---|
| | 10%BC | One-step | IQL | DT | RvS | POR (ours) | |
| antmaze-u | 62.8 | 64.3 | 87.5±2.6 | 59.2 | 65.4 | 90.6 ±7.1 | 89.0[BCQ] |
| antmaze-u-d | 50.2 | 60.7 | 66.2±13.8 | 53.0 | 60.9 | 71.3 ±12.1 | 61.0[BEAR] |
| antmaze-m-p | 5.4 | 0.3 | 71.2±7.3 | 0 | 58.1 | 84.6 ±5.6 | 68.0[CQL] |
| antmaze-m-d | 9.8 | 0 | 70.0±10.9 | 0 | 67.3 | 79.2 ±3.1 | 68.0[CQL] |
| antmaze-l-p | 0 | 0 | 39.6±5.8 | 0 | 32.4 | 58.0 ±12.4 | 18.8[CQL] |
| antmaze-l-d | 0 | 0 | 47.5±9.5 | 0 | 36.9 | 73.4 ±8.5 | 45.6[CQL] |
| antmaze mean | 21.4 | 20.8 | 63.6 ±8.3 | 18.7 | 53.5 | 76.2 ±8.1 | 58.4 |
| halfcheetah-r | 5.4 | 3.7±0.2 | 11.2±2.9 | - | - | 29 ±0.7 | 20.0[CQL] |
| hopper-r | 4.2 | 5.2±0.2 | 7.9±0.4 | - | - | 12±2.1 | 14.2 [BEAR] |
| walker2d-r | 6.7 | 5.6±0.6 | 5.9±0.5 | - | - | 6.3±0.3 | 8.3 [CQL] |
| halfcheetah-m | 42.5 | 48.4±0.1 | 47.4±0.2 | 42.6±0.1 | 41.6 | 48.8 ±0.5 | 48.3[TD3+BC] |
| hopper-m | 56.9 | 59.6±2.5 | 66.2±5.7 | 67.6±1.0 | 60.2 | 98.2 ±1.6 | 59.3[TD3+BC] |
| walker2d-m | 75.0 | 81.8±2.2 | 78.3±8.7 | 74.0±1.4 | 71.7 | 81.1±2.3 | 83.7 [TD3+BC] |
| halfcheetah-m-r | 40.6 | 38.1±1.3 | 44.2±1.2 | 36.6±0.8 | 38.0 | 43.5±0.9 | 45.5 [CQL] |
| hopper-m-r | 75.9 | 97.5±0.7 | 94.7±8.6 | 82.7±7.0 | 73.5 | 98.9 ±2.1 | 95.0[CQL] |
| walker2d-m-r | 62.5 | 49.5±12.0 | 73.8±7.1 | 66.6±3.0 | 60.6 | 76.6±6.9 | 81.8 [TD3+BC] |
| halfcheetah-m-e | 92.9 | 93.4±1.6 | 86.7±5.3 | 86.8±1.3 | 92.2 | 94.7 ±2.2 | 91.6[CQL] |
| hopper-m-e | 110.9 | 103.3±1.9 | 91.5±14.3 | 107.6 ±1.8 | 101.7 | 90.0±12.1 | 105.4[CQL] |
| walker2d-m-e | 109.0 | 113.0 ±0.4 | 109.6±1.0 | 108.1±0.2 | 106.0 | 109.1±0.7 | 110.1[TD3+BC] |
| locomotion mean | 55.5 | 58.2 ±2.0 | 57.6 ±6.6 | 56 ±1.4 | 53.7 | 65.6 ±2.7 | 63.5 |

chosen by the higher score between RvS-R and Rvs-G, which use the reward-to-go and **oracle** goal information as the conditioned variable, respectively. Note that we also categorize our method POR into conditioned BC methods because the execute-policy is learned in a similar manner to other conditioned BC methods. Note that the results of Filter BC are from our own implementation, and the results of IQL on MuJoCo random tasks are re-runed using the author-provided implementation. Other results are taken directly from their corresponding papers. To enable fairness and consistency in the evaluation process, we train our algorithm for 1 million time steps and evaluate every 5000 time steps that consist of 10 episodes. Full experimental details are included in Appendix D.1 and the learning curve can be found in Appendix E.

The results are shown in Table 1, in MuJoCo locomotion tasks, POR performs competitively to the best performance of prior methods in high-quality datasets, while achieving much better performance than other methods in low-quality datasets (e.g., `random` and `medium` datasets). On the more challenging AntMaze tasks, POR outperforms all other baselines by a large margin. It even surpassed RvS that uses the oracle goal information to learn the policy. We conjecture that the success in all those datasets should be credited to the out-of-distribution generalization ability of POR.

## 5.2 Validation Experiments on Out-of-Distribution Generalization

In this section, we try to validate our hypothesis about the out-of-distribution generalization of POR. We are especially interested to see how the guide-policy is learned in those high-dimensional tasks in D4RL. To do so, we compare the distribution of $g(s)$ generated by the guide-policy with the distribution of $s'$ from the offline dataset. We choose `antmaze-large-diverse` and `hopper-medium-replay`, two datasets for evaluation. For each dataset, we randomly collect $100,000$ $(s, s')$ transition tuples. To visualize the results clearly, we plot the distribution of $g(s)$ and $s'$ with t-Distributed Stochastic Neighbor Embedding (t-SNE) [18], we also visualize the true values of all states using the optimal value function $V^*(s)$.

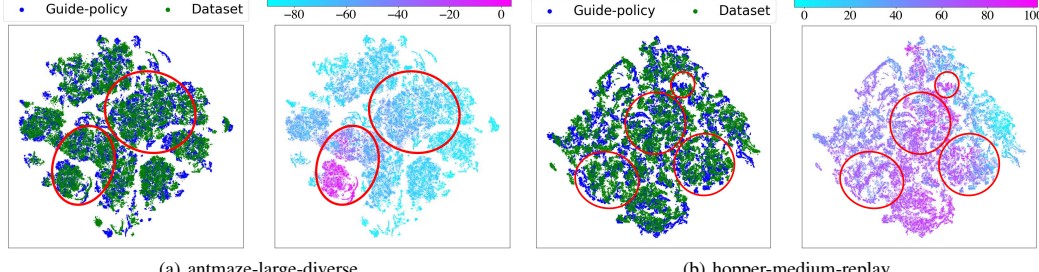

(a) antmaze-large-diverse

(b) hopper-medium-replay

Figure 2: Visualizations of the distribution of states generated by the guide-policy and states in the dataset, on `antmaze-large-diverse` and `hopper-medium-replay` datasets. The left panel of each subfigure shows the $100,000$ data points of $g(s)$ (blue) and $s'$ (green). The right panel of each subfigure shows the corresponding value of each state. The distribution of $g(s)$ and $s'$ bear some similarity but $g(s)$ is more prone to generate high-value states, especially on `antmaze-large-diverse` dataset. Better zoom in to see a more clear comparison.

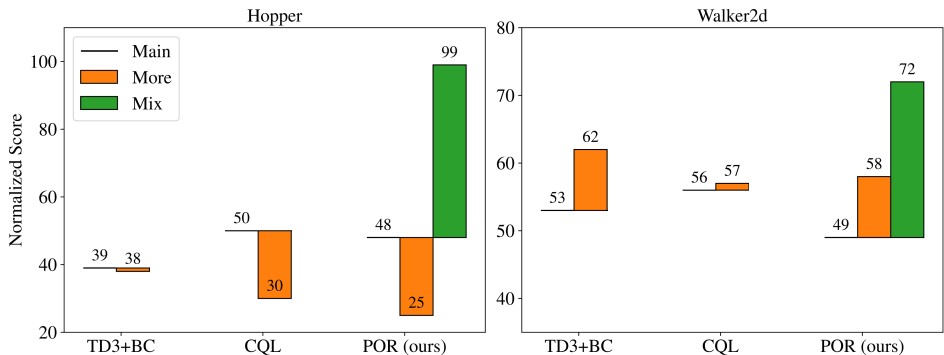

Figure 3: Normalize scores of different algorithms with different training schemes. While the "main" and "more" training scheme can be applied to all methods, the "mix" training scheme is only available to `POR` because we decouple the training process of $g$ and $\pi$. With the "mix" training scheme, `POR` outperforms all other algorithms by a large margin, in both `hopper` and `walker2d` environments.

The results are shown in Figure 2, it can be found that in both tasks, the distribution of $g(s)$ bears some similarity to the distribution of $s'$. However, when looking at the non-overlapped area of $g(s)$ and $s'$, we found that $g(s)$ has a much higher value than $s'$, especially in the `antmaze-large-diverse` dataset. Those out-of-distribution and high-value areas are exactly where `POR` generalized to, note that the high-value benefit of the single-step generalization could also be accumulated and propagated through time steps, resulting in a much better policy.

### 5.3 Investigations on the Guide-Policy

In this section, we investigate what is the potential benefit of decoupling the training process of the guide-policy and execute-policy. We found that as the execute-policy is not task-specific and only cares about the effect of actions on different states, one can reuse its powerful generalization ability when encountered with additional suboptimal data or adapting to a new task. We only need to re-train the guide-policy as it is task-specific.

**Improve guide-policy by additional suboptimal data** We first study the setting where we already have a policy learned from a small yet high-quality dataset $\mathcal{D}_e$, and now we get a supplementary large dataset $\mathcal{D}_o$. $\mathcal{D}_o$ may be sampled from one or multiple behavior policies, it provides higher data coverage but could be suboptimal. We are interested to see whether we can use $\mathcal{D}_o$ to obtain a better policy. This setting is realistic as low-quality datasets appear more often in real-world tasks [60, 57].

Naïvely, we can combine $\mathcal{D}_e$ and $\mathcal{D}_o$ as a new offline dataset $\mathcal{D}$, then run any off-the-shelf offline RL algorithm. We label this training scheme as "more". We also label the original training scheme that only uses $\mathcal{D}_e$ as "main". We test these two training schemes on two state-of-the-art offline RL algorithms, TD3+BC and CQL. The results shown in Figure 3 indicate that both "main" and "more"

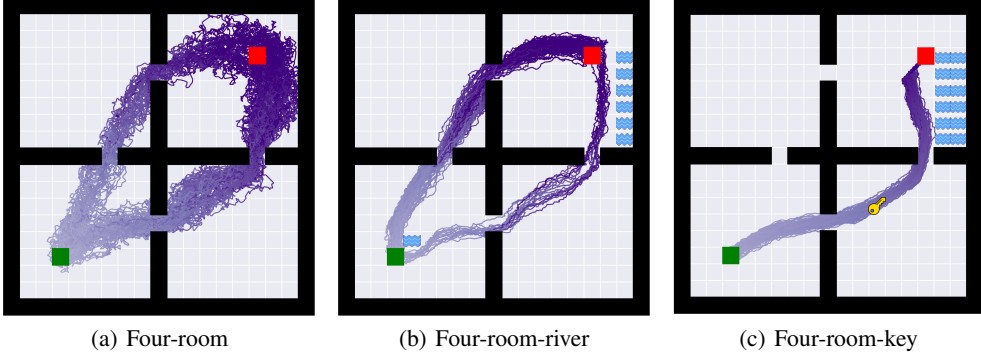

|  (a) Four-room | (b) Four-room-river | (c) Four-room-key |

Figure 4: Rollout trajectories of `POR` with the same pre-trained execute-policy but different guide policies in different four-room tasks, with color becoming more saturated as time progresses. The execute-policy is pre-trained in task A and remains unchanged in task B and task C. All three tasks require the agent to find a path from the start ■ to the goal ■. Besides, task B requires the agent not to fall into the river 〰 and task C requires the agent to get the key ⚷ as well as avoiding the river 〰 before arriving at the goal location.

are insufficient to learn a good policy, The reason why "main" fails is that the resulting policy may not be able to generalize due to the limited size. While "more" can generalize better than "main" due to access to a much larger dataset, the policy will be negatively impacted by the low-quality data in $\mathcal{D}_o$ as both TD3+BC and CQL contain the behavior regularization term.

Due to the availability of decoupled training, `POR` is able to use a different training scheme, we can keep the execute-policy learned from $\mathcal{D}_e$ unchanged but use the combination of $\mathcal{D}_e$ and $\mathcal{D}_o$ to re-train the guide-policy. Note that in this scheme, we allow the use of an *action-free* dataset $\mathcal{D}_o$, which often appears in real-world scenarios (i.e., video data and third-person demonstrations) [49]. The value function will be more accurate and better generalized when combining $\mathcal{D}_e$ with $\mathcal{D}_o$ to have a larger data coverage (see *exploration* in online RL as an example). We label this unique training procedure as "mix". It can be shown in Figure 3 that, by doing so, `POR` could have a super large improvement over "main" and "more". The version "mix" brings `POR` up to 99, close to the upper limit of the performance in the `hopper` environment, with only limited data. In the `walker2d` environment, `POR` with "mix" also outperforms all other baselines.

**Change guide-policy in new tasks**    Then, we study whether we can use the guide-policy to do a lightweight adaptation to new tasks, without changing the execute-policy. We conduct experiments on the continuous variant of the classic four-room environment [19]. The training data consists of trajectories collected by a goal-reaching controller with the start and end locations sampled randomly at non-wall locations, mixed with data collected by a random policy. In this environment, we introduce three different tasks: Task A (`Four-room`) needs the agent to travel from the start location to the goal location, besides that, Task B (`Four-room-river`) and task C (`Four-room-key`) require the agent to bypass the river or get a key before arriving at the goal location.

We first train the guide-policy and the execute-policy $\pi$ to solve task A. In task B and task C, we re-train the guide-policy using the corresponding reward function in that task and reuse the execute-policy $\pi$ from Task A. In Figure 4, we can see that by doing so, `POR` is able to successfully solve task B and task C. In detail, in task B, `POR` is aware of the river and correctly bypasses it. In task C, `POR` only chooses to pass the bottom door, this is owing to the awareness of the importance of obtaining the key before reaching the goal. This result shows the strong adaptation ablity of the guide-policy for different tasks, as well as the generalization ability of the execute-policy across different tasks.

# 6    Conclusions and Limitations

In this work, we propose a novel offline RL approach, `POR`, which leverages the training stability of imitation-style methods while still encouraging logical out-of-distribution generalization. `POR` allows *state-compositionality* rather than *action-compositionality* from the dataset. Through theoretical analysis and extensive experiments, we show that `POR` outperforms prior methods on a variety of

datasets, especially those with low quality. We also empirically demonstrate the additional benefits of POR in terms of improving with supplementary suboptimal data and easily adapting to new tasks. One limitation of POR is that the prediction error of the guide-policy may be large when the state space is high-dimensional. However, this can be alleviated with the help of representation learning [30, 64].

**Acknowledgments**    This work is supported by fundings from AsiaInfo. We thank Michael Janner for providing the code of the continuous four-room environment. We thank Junwu Xiong, James Zhang for their feedback on earlier versions of the manuscript.

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
