# A  More Discussion

**Why One-step and IQL are imitation-based methods?**    The core difference between RL-based and imitation-based methods is that RL-based methods learn a value function of policy $\pi$ while imitation-based methods don't. Learning the value function of $\pi$ requires off-policy evaluation of $\pi$ (i.e., learning $Q^\pi$ or $V^\pi$), which is prone to distribution shift. The policy evaluation and policy improvement will also affect each other as they are coupled.

Imitation-based methods don't learn $Q^\pi$ or $V^\pi$, but some of them do learn a value function. We call these methods imitation-based because they learn the value function using only dataset samples, the value function actually tells how advantageous it could be under the behavior policy, much like imitation learning. Also, the policy learning objective of One-step (with exponentially-weighted improvement operator) and IQL can be written as $L(\pi) = \mathbb{E}_{(s,a)\sim\mathcal{D}}\left[\exp\left(\beta\left(Q(s,a) - V(s)\right)\right)\log\pi(a|s)\right]$, which uses dataset actions to doing behavior cloning with different weights (One-step and IQL learn different value functions).

Cloning dataset actions can only do action-stitching, which loses the ability to surpass the dataset by out-of-distribution (action) generalization. For example, in our toy example in Section 4.1, action-stitching methods at most learn the shortest path that **contained** in the datset. This is suboptimal especially when there does not exist one complete path starting from the start location to the goal location in the offline dataset.

How can we do beyond action-stitching? One way is to use RL-based methods, by querying an accurate $Q^\pi$, we can get a different-yet-optimal action $a_\pi$, but $Q^\pi$ is hard to estimate. Another way is like what POR did, we learn an out-of-distribution state indicator, i.e., the guide-policy $g$, to guide the policy to the optimal next state. If the execute-policy can generalize well, it will also output a different yet optimal action $a = \arg\max_a \pi(a|s, g(s))$.

# B  More Related Work

Our work decouples the state-to-action policy into two modules, i.e., the guide-policy and the execute-policy. The execute-policy is actually an inverse dynamics model, which has been widely used in various ways in sequential decision-making. In exploration, inverse dynamics can be used to learn representations of the controllable aspects of the state [39]. In imitation learning, [49] and [33] train an inverse dynamics model to label the state-only demonstrations with inferred actions. [9] use inverse dynamics models to translate actions taken in a simulated environment to the real world.

Recently, there has been an emergence of work [45, 15] highlighting the connection of imitation learning and reinforcement learning. Specifically, rather than learn to map states and actions to reward, as is typical in reinforcement learning, [45] trains a model to predict actions given a state and an outcome, which could be the amount of reward the agent is to collect within a certain amount of time. [15] uses a similar idea, predicting actions conditioned on an initial state, a goal state, and the amount of time left to achieve the goal. These methods are perhaps the closest work to our algorithm, however, we study the offline setting and motivate the usage of an inverse dynamics model from a different perspective (i.e., state-stitching).

# C  Proof

In this section, we provide the proof of Theorem 1.

*Proof.* We can rewrite the LHS of Eq.(8) as

$$
\begin{aligned}
\|\pi(s, g(s)) - a^*\| &= \|\pi(s, g(s)) - \pi(s, s') + \pi(s, s') - a + a - a_g + a_g - a^*\| \\
&\leq \|\pi(s, g(s)) - \pi(s, s')\| + \|\pi(s, s') - a\| + \|a - a_g\| + \|a_g - a^*\| \ \ \text{(Triangle)} \\
&\leq L_2\|g(s) - s'\| + \epsilon + L_1\|g(s) - s'\| + \|a_g - a^*\| \ \ \text{(Assumption1\&2)} \\
&\leq \underbrace{(L_1 + L_2)\|g(s) - s'\|}_{l_1} + \underbrace{\|a_g - a^*\|}_{l_2} + \underbrace{\epsilon}_{l_3}
\end{aligned}
$$

$\square$

# D   Experimental Details

In this section, we provide the experimental details of our paper. We use the following hardware and software for our training:

- GPUs: NVIDIA GeForce RTX 3080Ti
- Python 3.7
- Pytorch 1.10.0
- Gym 0.23.1 [6]
- MuJoCo 2.1.4 [48]
- mujoco-py 2.1.2.14

## D.1   D4RL Experiments

**Data collection**   The datasets in D4RL have been generated as follows: `random`: roll out a randomly initialized policy for 1M steps. `expert`: 1M samples from a policy trained to completion with SAC [17]. `medium`: 1M samples from a policy trained to approximately 1/3 the performance of the expert. `medium-replay`: replay buffer of a policy trained up to the performance of the medium agent. `medium-expert`: 50-50 split of medium and expert data. For all datasets we use the v2 version.

**Implementation details**   Our implementation of 10%BC is as follows, we first filter the top 10 % trajectories in terms of the trajectory return, and then run behaviour cloning on those filtered data. The hyperparameters of `POR` are present in Table 2. We use target networks for the $V$-function and use clipped double $V$-learning (take the minimum of two $V$-functions) for all updates. We normalize state to $[-1, 1]$ [42] to reduce the prediction error of the guide-policy, it can be deemed as an naïve method of representation learning.

Table 2: `POR` Hyperparameters.

|  | Hyperparameter | Value |
|---|---|---|
|  | Value network hidden dim | 256 |
|  | Value network hidden layers | 2 |
|  | Value network activation function | ReLU |
|  | Guide-policy hidden dim | 256 |
| Architecture | Guide-policy hidden layers | 2 |
|  | Guide-policy activation function | ReLU |
|  | Execute-policy hidden dim | 512 |
|  | Execute-policy hidden layers | 2 |
|  | Execute-policy activation function | ReLU |
|  | Optimizer | Adam [23] |
|  | Value netowrk learning rate | 3e-4 |
|  | Target Value netowrk moving average | 0.05 |
|  | Guide-policy learning rate | 1e-3 |
| POR Hyperparameters. | Execute-policy learning rate | 1e-3 |
|  | Mini-batch size | 256 |
|  | Discount factor | 0.99 |
|  | $\tau$ | 0.7 (Mujoco), 0.9 (AntMaze) |

## D.2   Additional Suboptimal Data Experiments

**Data collection and settings**

In this experiment, we use different part of `medium-replay` datasets as $\mathcal{D}_e$ and $\mathcal{D}_o$. More specific, we use 30% to 70% transitions to constitute $\mathcal{D}_e$ and use 20% to 80% transitions to constitute $\mathcal{D}_e \cup \mathcal{D}_o$.

**Implementation details** We use the same hyperparamters of POR, shown in Table 2. Our implementations of TD3+BC[†] [14], CQL[‡] [28] is from the author-provided implementation from Github, and we keep all parameters the same to the author-provided implementation.

Table 3: The hyperparameters of CQL in additional-data experiments.

|  | Hyperparameter | Value |
|---|---|---|
| Architecture | Critic hidden dim | 256 |
|  | Critic hidden layers | 3 |
|  | Critic activation function | ReLU |
|  | Actor hidden dim | 256 |
|  | Actor hidden layers | 3 |
|  | Actor activation function | ReLU |
| CQL Hyperparameters | Optimizer | Adam [23] |
|  | Critic learning rate | 3e-4 |
|  | Actor learning rate | 1e-4 |
|  | Mini-batch size | 256 |
|  | Discount factor | 0.99 |
|  | Target update rate | 5e-3 |
|  | Target entropy | $-1 \cdot$ Action Dim |
|  | Entropy in Q target | True |
|  | Lagrange | False |
|  | Num sampled actions (during eval) | 10 |
|  | Num sampled actions (logsumexp) | 10 |
|  | $\alpha$ | 10 |

Table 4: The hyperparameters of TD3+BC in additional-data experiments.

|  | Hyperparameter | Value |
|---|---|---|
| Architecture | Critic hidden dim | 256 |
|  | Critic hidden layers | 2 |
|  | Critic activation function | ReLU |
|  | Actor hidden dim | 256 |
|  | Actor hidden layers | 2 |
|  | Actor activation function | ReLU |
| TD3+BC Hyperparameters | Optimizer | Adam [23] |
|  | Critic learning rate | 3e-4 |
|  | Actor learning rate | 3e-4 |
|  | Mini-batch size | 256 |
|  | Discount factor | 0.99 |
|  | Target update rate | 5e-3 |
|  | Policy noise | 0.2 |
|  | Policy noise clipping | (-0.5, 0.5) |
|  | Policy update frequency | 2 |
|  | $\alpha$ | 2.5 |

### D.3 Four-room Experiments

**Environment settings** We use the continuous variant of the classic four-room environment from [19]. This continuous variant of four-rooms is basically the same as the traditional classic four-room environment in the environment design. There are $19 \times 19$ grids that consist of four rooms with only one block channel with neighboring rooms. The goal of this environment is to make the agent

---

[†]https://github.com/sfujim/TD3_BC
[‡]https://github.com/aviralkumar2907/CQL

travel from one location to another different location. Those tasks are challenging as their reward is extremely spare, they need the agent to have the ability to explore efficiently through the whole state space. The action and observation space are shown in Table 5. The first and second dimension of the action space represents the distance to travel on the $x$-axis and $y$-axis, respectively. The first and second dimension of the observation space represents the coordinate on the $x$-axis and $y$-axis, respectively.

Table 5: The action and observation space in the continuous `Four-room` environment.

| | |
|---|---|
| Action space | Box(-0.1, 0.1, (2,), float32) |
| Observation space | Box(-18, 18, (2,), float32) |
| Action dimension | 2 |
| Observation dimension | 2 |

In task A (`Four-room`), only reaching the goal will give the agent a reward of 1, otherwise the agent will get 0 reward. When the agent reaches the target or the agent takes over 500 steps, the environment will terminate. In task B (`Four-room-river`), the agent will receive $-1$ reward if it falls into the river, and the environment will be terminated. Also, the agent will get 1 reward only when it reaches the goal. In task C (`Four-room-key`), the agent gets 1 reward only when it reaches the goal **and** gets the key. Falling into the river will also give the agent $-1$ reward.

**Data collection and implementation details** The training data consists of trajectories collected by a goal-reaching controller with the start and end locations sampled randomly at non-wall locations. To make the data more diverse, we also add trajectories from a random policy. We collect 100,000 transitions for each task. Figure 6 shows the training hyperparameters of `POR` in the three four-room environments.

Table 6: The hyperparameters of `POR` in continuous `Four-room` environments.

| | Hyperparameter | Value |
|---|---|---|
| | Value network hidden dim | 64 |
| | Value network hidden layers | 2 |
| | Value network activation function | ReLU |
| | Guide-policy hidden dim | 64 |
| Architecture | Guide-policy hidden layers | 2 |
| | Guide-policy activation function | ReLU |
| | Execute-policy hidden dim | 64 |
| | Execute-policy hidden layers | 2 |
| | Execute-policy activation function | ReLU |
| | Optimizer | Adam [23] |
| | Value network learning rate | 3e-4 |
| | Target V moving average | 0.05 |
| | Guide-policy learning rate | 1e-3 |
| | Execute-policy learning rate | 1e-3 |
| Training Hyperparameters | Mini-batch size | 256 |
| | Discount factor | 0.99 |
| | Normalize | False |
| | $\tau$ | 0.9 |

# E    Learning Curves

In this section, we provide the learning curves of our experiments in the main paper.

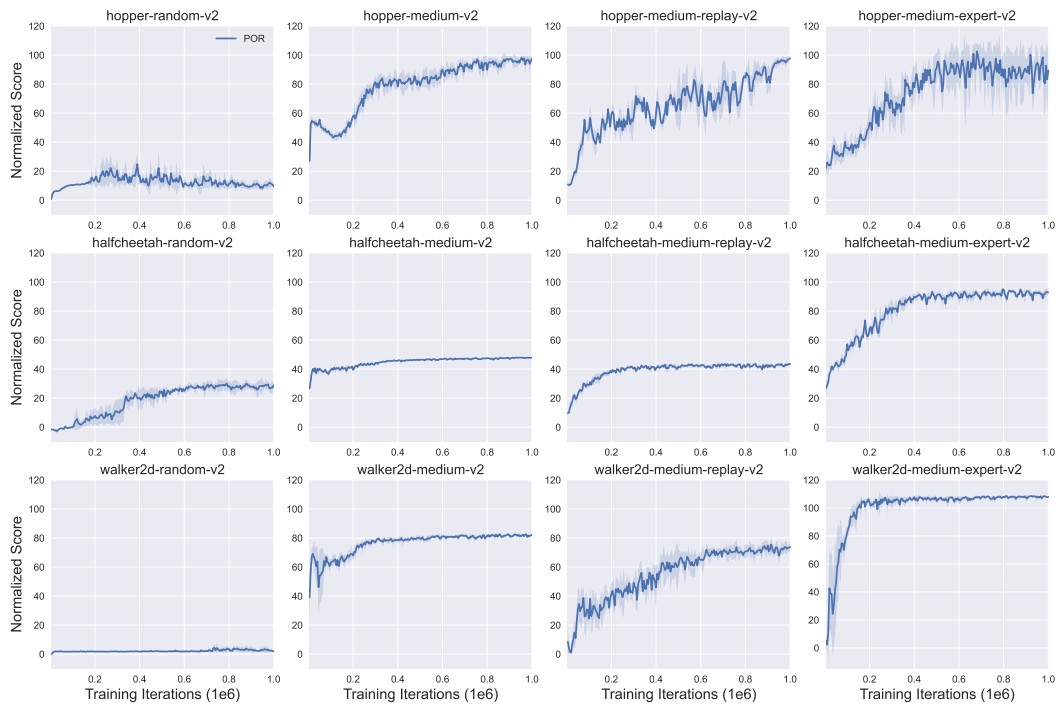

Figure 5: Learning curves on MuJoCo locomotion datasets.

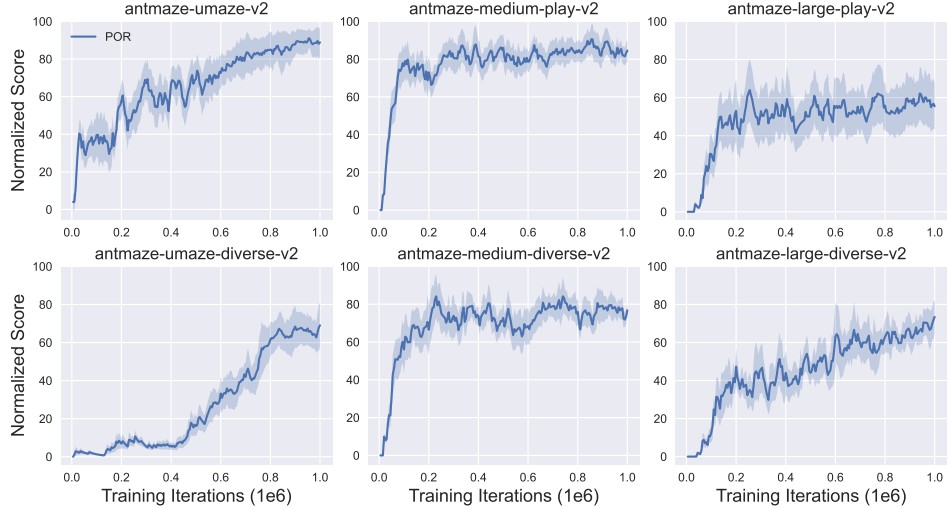

Figure 6: Learning curves on D4RL Antmaze datasets.

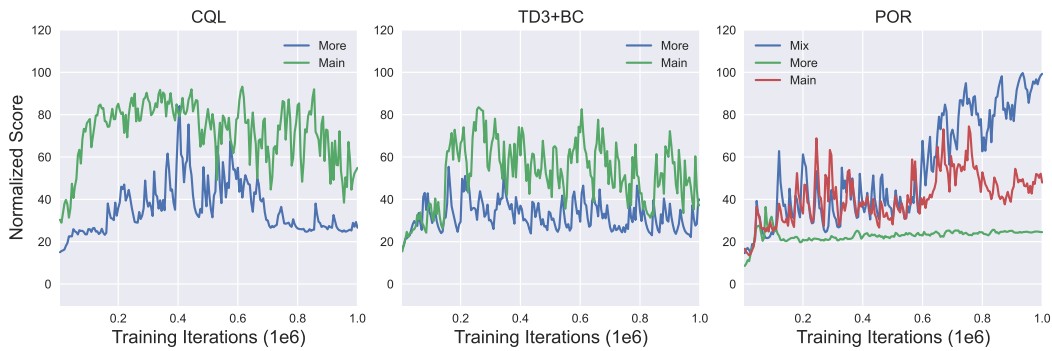

Figure 7: Learning curves of additional-data experiments on hopper-medium-replay-v2 datasets.

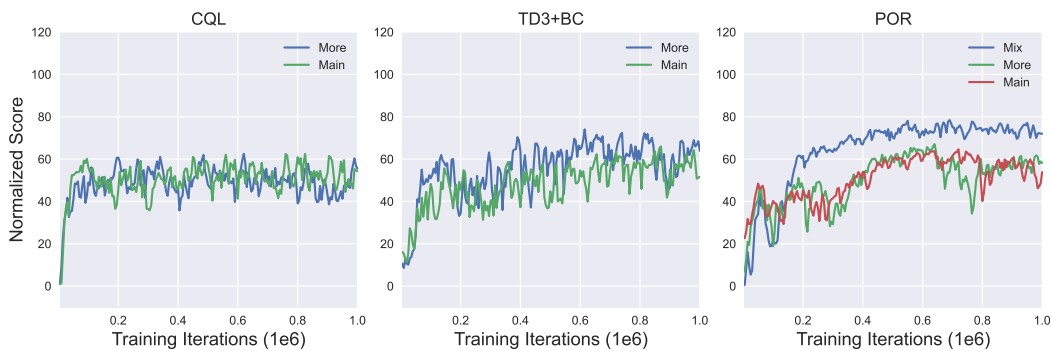

Figure 8: Learning curves of additional-data experiments on walker2d-medium-replay-v2 datasets.

## F  Ablation Study on the Execute-Policy

We also present an ablation study on the execute-policy, with the aim to answer the following two questions: 1) How does network capacity affect the performance of the execute-policy? 2) How does the performance change if we use partial "good" data, instead of full data, to train the execute policy?

To answer the first question, we compare two choices of network size: a big network with $(512, 512)$ hidden units and a small network with $(128, 128)$ hidden units. To answer the second question, we compare with the choice that only selects a subset of the dataset to train the execute policy. Concretely, we choose the top $X\%$ trajectories in the dataset, ordered by episode returns. We sweep $X$ over $[10, 25, 40]$ and choose the best score. We give the mean scores of AntMaze (`A`) and MuJoCo (`M`) datasets in Table 7. It can be seen that adopting a big network consistently gives a better performance, which is also found in [10]. Using partial data will result in a less-performed policy, especially on AntMaze datasets. Note that in MuJoCo datasets, the performance didn't drop too much. This is because MuJoCo datasets don't require the compositionality ability, using partial trajectories could already achieve high scores.

Table 7: Ablation study of the execute-policy on network capacity and dataset size.

|  | Big Network | Small Network |
|---|---|---|
| Full $\mathcal{D}$ | `A`: 76.2, `M`: 65.7 | `A`: 63.8, `M`: 57.0 |
| Partial $\mathcal{D}$ | `A`: 52.4, `M`: 60.9 | `A`: 41.5, `M`: 56.7 |