# OpenReview forum: "A Policy-Guided Imitation Approach for Offline Reinforcement Learning"
_NeurIPS.cc/2022/Conference — NeurIPS 2022 Accept_

### Official Review · Reviewer_ED7f · 2022-07-02

**Rating:** 7
**Confidence:** 3
**Soundness:** 4 excellent
**Presentation:** 3 good
**Contribution:** 3 good

**Summary:**

This paper proposes a totally new paradigm to achieve Offline RL, compared to the alternatives that stitch actions for constraining the target policy to stay close to the behavior policy. Instead of composing actions, the proposed $POR$ allows state compositionality through learning a guided policy $g(s)$ producing **target state to reach** and optimizing the execute policy $\pi(a|s, s')$ conditioned on the goal state in an imitation learning manner. After that, the agent makes decisions according to $\underset{a}{\mathrm{argmax}}\ \pi(a|s, g(s))$ in the inference stage.

The decoupled two-stage learning process benefits from the Imitation-based method's stability and the RL-based method's out-of-distribution generalizability. Theoretical analysis shows the lower bound of the proposed method and how this bound will be influenced by the guided policy $g(s)$. The comprehensive experiments show that $POR$ achieves SOTA on *Mujoco* and *AntMaze* dataset and new applications are enabled by this new learning paradigm.

**Questions:**

Q1: As stated in the paper, $POR$ can pick logical out-of-distribution actions for reaching the state-to-go produced by $g(s)$. However, in my view,  although the $g(s)$ can generate an out-of-distribution goal state $s'$ for $\pi$, the lack of transition $(s, a, s')$ may prevent $\pi$ from arriving $s'$. Let's consider the toy example. When the agent is at $s=(2, 0)$ and the $\hat{s}=g(s)$ indicates that $\hat{s}=(3, 1)$ has the highest value, the policy is supposed to select the upper right action according to $\hat{a}=\underset{a}{\mathrm{argmax}}\ \pi(a|s, \hat{s})$. However, there is no such a transition $(s, \hat{a}, \hat{s})$ contained in the dataset, and hence the training policy $\pi$ can not know how to arrive $(3, 1)$. Would the policy $\pi$ fails to reach the target state $\hat{s}$ in this situation?

**Limitations:**

I am wondering whether this method can still work well on discrete control tasks.

**Strengths And Weaknesses:**

Strength:
1) It is an interesting idea to design a goal-conditioned policy for Offline RL tasks. The learned policy only encodes state reachability via supervised learning and thus can learn stably. In addition, the out-of-distribution generalization is completed through stitching states via a variant of *Bellman Operator* and yields $s'=g(s)$ for providing a high-value target goal $s'$.
2) The paper is well.
3) Authors do comprehensive experiments to evaluate the new method and discuss its potential applications in the real world.

Weakness and typos:
1) Action selection should follow **Eq.(6)** in Algorithm 1, line 16 :).
2) The axes can be added to Figure. 1 showing the toy example, so readers can know if the start index is 0 or 1. It indeed takes some time for me to find (5,4) and (6,5)
3) The results of t-NSE are not clear enough, especially in Fig. 2, left panel, ant-maze-large. It only shows that the guided policy $g(s)$ can select some out-of-distribution goal states. However, it is hard to observe that these out-of-distribution states are in higher value than the states contained in the dataset. Consider using circles in the figure to highlight which area shows the $g(s)$ manages to generate high-value OOD states to help understand this figure.

---

> ### Author Response · Authors · 2022-08-02
> **Response to Reviewer ED7f**
>
> We thank the reviewer for the thorough and detailed comments.
>
> Regarding the weakness and typos you posted: we have fixed the typos and revised the pictures within the paper to reflect your concerns and suggestions, please see our revision.
>
> Regarding your questions: yes, you are right. The learned execute-policy enables generating correct actions to reach the desired state via the generalization ability of neural networks. It may fail to generalize due to limited data size or narrow data distribution, we theoretically give the generalization bound with respect to the guide-policy in Theorem 1, and we show empirically that by using a proper guide-policy, the execute-policy can achieve good results.
>
> Regarding the limitations you posted, we think our method can still work on discrete control tasks. The execute-policy may be easier to learn if the action only has several choices. We will consider adding some discrete control tasks in the latter revision.

---

> > ### Comment · Reviewer_ED7f · 2022-08-08
> > **Reviewer Response**
> >
> > Thanks for the response. I will keep my score unchanged

---

### Official Review · Reviewer_iPbB · 2022-07-08

**Rating:** 8
**Confidence:** 4
**Soundness:** 4 excellent
**Presentation:** 3 good
**Contribution:** 4 excellent

**Summary:**

This paper proposes an approach to offline RL that first learns a policy $\pi(s' \mid s)$ for predicting the next *state*, and then learns another policy for extracting the corresponding action. The next-state policy ("guide policy") is trained to maximize a learned value function while staying close to the next states in the dataset. The action policy ("execution policy") is trained using a standard inverse action loss. Empirically, the proposed method outperforms prior methods on many benchmark tasks, especially ones involving sub-optimal data. Additional experiments show that the method affords efficient transfer to new tasks by finetuning just one of the two policies.

--------------------
**After author response**: The authors did a great job addressing my questions and concerns. I continue to think that this is a strong paper and should be accepted.

**Questions:**

1. Since the guide policy is predicting the next state, it's output could be very high dimensional, right? Does this pose a problem?
2. How does network capacity affect the performance of the *guide* policy?
3. Eq. 4 -- How carefully does $\alpha$ have to be tuned?
4. For the experiments in Fig 3, I suspect that most of the benefit from the "mixed" approach is that learning the execute policy on the expert data effective amounts to behavioral cloning (e.g., see analysis in [3]). If so, this benefit can be injected into the TD3+BC and CQL methods, too. For example, for TD3+BC, we could learn the Q-function on the combined dataset and train the actor on just the expert data. *How would these "mixed" versions of TD3+BC and CQL perform?*
4. Can the method be used to perform transfer across agents with different action spaces? E.g., learn a guide policy for one robot and an execute policy for another robot, and then mix and match them.


[3] https://arxiv.org/pdf/2206.03378v1.pdf, Section 3.2

**Limitations:**

Yes, the limitations are well addressed.

**Strengths And Weaknesses:**

Strengths
* The idea is novel and makes intuitive sense.
* The experimental results are strong, with a lot of care put into making them reproducible.
* The additional experiments were good for building intuition into the method.
* Fig 1 is great for building intuition (though the description could be clarified).
* It's great that the paper provides theoretical results relating the policy learning losses to the performance of the resulting policy.

Weaknesses
* It's not entirely clear *why* the proposed method should work, especially since the guide policy's job of predicting the next state seems  onerous in high-dimensional tasks (e.g., AntMaze).
* The writing is legible, but grammatical and spelling errors are frequent.


Minor writing comments:
* Abstract -- I didn't understand the method after reading the abstract (but the last two paragraphs of the intro do provide a good succinct description)
* L12 "Prophet" -- I found this pretty confusing, as it was unclear how this relates to the execute and guide policies. I might recommend renaming everything to refer to its inputs and outputs (e.g., inverse action policy, next state policy)
* L16 "we highlight" -- Great contributions!
* L25 "requires" --> "often entails" (this isn't a strict requirement)
* L29 "can not ..." -- Cite.
* Introduction -- At some point, it would be useful to more formally characterize the difference between the IL and RL methods being discussed (e.g., "We use imitation learning to refer to methods that do not learn a value function.")
* L49 "optimal next state ... " -- Great explanation.
* L55 "which owns logical" -- I didn't understand this.
* L60 "We also show ..." -- Great description of the contributions.
* L64 "," --> "."
* L140 "second method" -- Does this second method implicitly assume access to an oracle execute policy? If the action has never been taken before, then how would the agent know which action to take?
* L141 "acction" --> "action"
* L164 "orronumously" --> "erroneously"
* L170 -- L179 -- It would be good to clarify that none of this material is novel, but instead is taken from IQL. This will help the reader distinguish the contributions of the paper.
* Sec. 4.3 -- Cite many other papers that use inverse action models [1, 2], which are more relevant than RvS.
* L206 "conditioning variable can be automatically generated" -- Great point.
* Assumption 1 -- Is $(s_1, s_1')$ treated as a big, concatenated vector when taking the norm? Which norm is this?
* Assumption 2 -- It seems like this can be made to hold by simply restricting the policy class. That is, this is not an assumption on the problem, and instead could be treated as part of the algorithm.
* L224 -- L229, "meaningless", "in vain", "plenty of worls" -- Too colloquial for technical writing.
* Eq 7 -- Precisely, how is $a^*$ defined?
* Theorem 1 -- Does this suggest that $\pi$ should be trained with the $\ell-\infty$ norm used for $\epsilon$, or that $\alpha = L_1 + L_2$?
* Theorem 1 -- It would be good to acknowledge the limitation that computing the $\ell-\infty$ norm for $\epsilon$ is quite hard, and its value will be quite large, potentially making the bound loose.
* L253 "benfits" --> "benefits"
* L255 "on D4RL" --> "on the D4RL"
* L256 "are consisted of" --> "consist of"
* (many places) -- "Imitation" --> "imitation"
* L267 -- L275 -- Great job making the results reproducible.
* Fig 2 -- I was unsure how to interpret the right panel of these plots. It seems like the higher values are in the bottom/left, but I'm unsure how that relates to $g(s)$ and $s'$.
* L308 "this two approach as" --> "these two approaches as"
* L318 "lable" --> "label"
* L320 "up tp" --> "up to"
* L361 "may be biased" -- I didn't understand this.

[1] https://arxiv.org/pdf/1912.12773.pdf

[2] http://proceedings.mlr.press/v70/pathak17a/pathak17a.pdf

---

> ### Author Response · Authors · 2022-08-02
> **Response to Reviewer iPbB**
>
> Thank you very much for your kind suggestions and detailed review, you are one of the most responsible reviewers we have encountered!
>
> We especially appreciate for minor writing comments you posted, we have made several writing adjustments within the paper to reflect some of your concerns and suggestions, please see our revision. There are still some we haven't solved, we will definitely solve it all in the latter version.
>
> >"Since the guide policy is predicting the next state, it's output could be very high dimensional, right? Does this pose a problem?"
>
> Yes, we agree that this is one weakness of POR. Fortunately, the state space of antmaze is 29, it is not super large, and we may need to use representation learning to reduce the dimension in those high-dimensional tasks.
>
> >"How does network capacity affect the performance of the guide policy?"
>
> We compare the used guide-policy network size ($(256, 256)$ in the paper) to a smaller one $(128, 128)$ and a larger one $(1024, 1024)$, and the results are listed as follows. We found that using a small network size is insufficient for learning in some datasets but a much larger network size didn't guarantee higher scores.
>
> | Dataset      | (128, 128) | (256, 256) | (1024, 1024) |
> | ----------- | ----------- | ----------- | ----------- |
> | antmaze-m-p     | 64.6+7.8 | 84.6+5.6 | 80.2+3.3 |
> | antmaze-m-d     | 70.5+9.2 | 79.2+3.1 | 80.4+6.5 |
> | hopper-m-r      | 40.6+14.2 | 98.9+1.6 | 99.5+3.2 |
> | walker2d-m-r    | 36.4+15.2 | 76.6+6.9 | 76.4+7.2 |
>
>
> >"Eq. 4 -- How carefully does $\alpha$ have to be tuned?"
>
> We set it to 0.2 in AntMaze tasks and 0.5 in MuJoCo tasks.
>
> >"For the experiments in Fig 3, I suspect that most of the benefit from the "mixed" approach is that learning the execute policy on the expert data effective amounts to behavioral cloning (e.g., see analysis in [3]). If so, this benefit can be injected into the TD3+BC and CQL methods, too. For example, for TD3+BC, we could learn the Q-function on the combined dataset and train the actor on just the expert data. How would these "mixed" versions of TD3+BC and CQL perform?"
>
> This is a good question. First, we want to apologize that we didn't make the description of the setting in Section 5.3 clear enough, and we have revised the paper to make it more clear. We want to study the setting where we already have a policy learned from a small yet high-quality dataset $\mathcal{D}_e$, and now we get a supplementary large dataset $\mathcal{D}_o$. $\mathcal{D}_o$ may be sampled from one or multiple behavior policies, it provides higher data coverage but could be suboptimal. We are interested to see whether we can use $\mathcal{D}_o$ to obtain a better policy.
>
> So in this setting, only POR is able to keep the execute-policy learned from $\mathcal{D}_e$ unchanged but use the combination of $\mathcal{D}_e$ and $\mathcal{D}_o$ to re-train the guide-policy (i.e., the "mixed" training scheme) due to the decoupled training procedure, that's the reason why initially we didn't include the "mixed" version of TD3BC and CQL. After reading your review, we immediately try the "mixed" version of TD3BC and CQL, the results are listed as follows.
>
> | Dataset                   | TD3BC-main | TD3BC-more | TD3BC-mixed | CQL-main | CQL-more | CQL-mixed |
> | -----------               | -----------| -----------| -----------|----------| ---------|  ---------|
> | hopper-medium-replay-v2   | 39.1   | 38.7   | 37.5   | 50.5 | 30.1 | 55.6  |
> | walker2d-medium-replay-v2 | 53.6   | 62.4   | 69.7   | 56.8 | 57.4 | 66.1  |
>
>
> The results show that the "mixed" version of TD3BC and CQL perform inferior to the "mixed" version of POR. We have not figured out why, one reason we suspect is that when computing the target $Q$ value, the policy may generate some OOD actions since $Q^{\pi}$ is trained on $\mathcal{D}_e$ and $\mathcal{D}_o$ while the policy is trained only on $\mathcal{D}_e$, the actions produced by the policy may have a large discrepancy from actions in $\mathcal{D}_o$, causing large policy evaluation errors.
>
>
> >"Can the method be used to perform transfer across agents with different action spaces? E.g., learn a guide policy for one robot and an execute policy for another robot, and then mix and match them."
>
> Thanks for pointing out that! A really good point. We think our method could do this since the guide-policy only indicates which state is more optimal, we will add some experiments to verify that in the latter revision.

---

> > ### Comment · Reviewer_iPbB · 2022-08-05
> > **Reviewer response**
> >
> > Thanks for the answers/comments! I continue to think that this is a good paper.

---

### Official Review · Reviewer_PSxU · 2022-07-10

**Rating:** 6
**Confidence:** 4
**Soundness:** 2 fair
**Presentation:** 3 good
**Contribution:** 3 good

**Summary:**

The paper studies offline reinforcement learning (RL) and proposes to decouple the reward-maximizing objective in RL into learning a guide-policy and an execute-policy. For learning the guide-policy, the authors adopt the expectile regression and an additional BC loss. For learning the execute-policy, the author performs supervised learning by maximizing a likelihood. During the test, the guide-policy tells the execute-policy where it should go, and then the execute-policy conditioned on the current state and goal-state outputs the action. The authors also show the effectiveness of the method on a range of tasks (including the standard D4RL tasks and some transfer tasks).

**Questions:**

1. What is the direct benefit of using expectile regression to learn g_\omega rather than \pi(a|s). The authors state that "we only need to learn a value function while IQL needs an additional Q-function to extract the policy". However, the authors also introduce an additional guide-policy, as well as the execute-policy.
2. Intuitively, the expectile regression can sometimes eliminate the OOD issue in offline RL, but the authors additional introduce a behavior cloning term in Equation 4. My question is, if we replace the expectile regression in Equation 3 with the standard regression loss and then adopt the behavior cloning loss in Equation 4, what will be the difference in performance. More ablation experiments are required.
3. The authors use both expectile regression and BC loss for learning the guide-policy.  Doesn't this make learning too conservative?
4. In essence, the proposed method requires learning addition execute-policy. I do not see why there would be an added advantage over IQL on the D4RL tasks.
5. In Section 4.1, the authors state that "to generate better-than-dataset trajectories, the agent needs to take logical out-of-distribution actions". However, the toy example assumes that offline data includes transitions of taking random actions at random states. So, is there any OOD action in this motivating example? Further, does the learned execute-policy enable the output of the so-called "logical" out-of-distribution actions?
6. In Equation 4, are sampled states s and s' at two adjacent time steps? If so, I think the execute-policy will not output the (logical) out-of-distribution actions, given the current state s and goal-state s' are two adjacent states.

**Limitations:**

yes

**Strengths And Weaknesses:**

Strengths:
+ The presented method is simple and well-motivated.
+ The paper is overall well-executed.
+ Experiments cover various scenarios, showing the effectiveness on D4RL tasks and the generalization ability of the execute-policy.

Weaknesses:
+ The proposed method is somewhat incremental, which combines the expectile regression (and a BC loss in Equation 4) for eliminating the OOD issue in offline setting and supervised goal-reaching execute-policy learning.
+ The authors adopt the expectile regression from IQL, but do not explain the direct benefits of using expectile regression to learn g_\omega rather than \pi(a|s).
+ Some references are misquoted, eg. seeing Line 264.
+ The description of some references is inaccurate. For example, the authors state that One-step RL [3] and IQL use different weights to do behavior cloning. Is that true?
+ The representation of some formulas is not rigorous enough. For example, a_0 in Equations 1 should be a_0 \sim \pi(a_0|s_0). a_t and s_t in Equation 2 should be given a clear description, eg. the relation to \tau.

---

> ### Author Response · Authors · 2022-08-02
> **Response to Reviewer PSxU**
>
> We thank the reviewer for the thorough and detailed comments.
>
> >"The description of some references is inaccurate. For example, the authors state that One-step RL [3] and IQL use different weights to do behavior cloning. Is that true?"
>
> We have included a whole section to discuss this, please see Appendix A for details.
>
> >"What is the direct benefit of using expectile regression to learn g_\omega rather than \pi(a|s)."
>
> If we understand correctly, the question you want to ask is that "what is the benefit of using expectile regression to learn the state value function in POR against to learn the state-action value function in IQL?".
>
> The high-level idea of IQL and POR is different in that IQL is doing action-stitching while POR is doing state-stitching.
>
> IQL uses expectile regression to approximate the expectile of $Q$ with respect to the action distribution given a state. To extract the policy, IQL maximizes this objective, $L(\pi)=\mathbb{E}_{(s, a) \sim \mathcal{D}}\left[\exp \left(\beta\left(Q(s, a)-V(s)\right)\right) \log \pi(a|s)\right]$, which can be deemed as using dataset actions to do weighted behavior cloning.
>
> Cloning dataset actions can only do action-stitching, which loses the ability to surpass the dataset by out-of-distribution (action) generalization. In POR, we learn an out-of-distribution state indicator, i.e., the guide-policy $g$, to guide the policy to the optimal next state. If the execute-policy can generalize well, it will output a different yet optimal action $a=\arg \max_{a} \pi(a | s, g(s))$.
>
> >"The authors state that "we only need to learn a value function while IQL needs an additional Q-function to extract the policy"."
>
> We apologize for the misleading statement here. We want to discuss the difference between learning $V$ in POR and learning $Q$ in IQL, not to mean that the benefit of POR is only need to learn one value function rather than two. We have revised our statement in the paper to make it more clear.
>
>
> >"Intuitively, the expectile regression can sometimes eliminate the OOD issue in offline RL, but the authors additional introduce a behavior cloning term in Equation 4. My question is, if we replace the expectile regression in Equation 3 with the standard regression loss and then adopt the behavior cloning loss in Equation 4, what will be the difference in performance. More ablation experiments are required."
>
> The expectile regression technique and behavior cloning loss are two orthogonal term. Yes, expectile regression can eliminate the OOD issue because it uses only dataset samples, but only when we are learning the value function. When we use the value function to extract the policy, it will still query the value function about actions (states in our case) that generated by the policy, which are out of distribution. So we need the behavior cloning loss in Equation 4.
>
> We add a ablation study that replaces the expectile regression in Equation 3 with the standard mse loss. The results are listed as follows, as expected, the standard mse loss performs worse than the expectile regression loss.
>
> | Dataset      | Expectile regression | Mean-squared error |
> | ----------- | ----------- | ----------- |
> | antmaze-m-p     | 84.6$\pm$5.6 | 25.0$\pm$18.4 |
> | antmaze-l-p     | 58.0$\pm$12.4 | 15.2$\pm$7.0 |
> | hopper-m-r      | 98.9$\pm$2.1 | 30.7$\pm$2.4 |
> | walker2d-m-r    | 76.6$\pm$6.9 | 70.5$\pm$3.4 |
>
>
> >"The authors use both expectile regression and BC loss for learning the guide-policy. Doesn't this make learning too conservative?"
>
> As we discussed above, these two term are orthogonal, it won't make learning too conservative.
>
> >"In essence, the proposed method requires learning addition execute-policy. I do not see why there would be an added advantage over IQL on the D4RL tasks."
>
> As I discussed above, POR is doing state-stitching rather than action-stitching, which reduces conservatism and allows more out-of-distribution generalization.
>
> >"However, the toy example assumes that offline data includes transitions of taking random actions at random states. So, is there any OOD action in this motivating example?"
>
> In the motivating example, the dataset is colored green, we do not generate random actions at every state, so there are plenty of OOD actions.
>
> >"Further, does the learned execute-policy enable the output of the so-called "logical" out-of-distribution actions?"
>
> This is a good question. The learned execute-policy enables generating correct actions to reach the desired state via the generalization ability of neural networks. It may fail to generalize due to limited data size or narrow data distribution, we theoritically give the generalization bound with respect to the guide-policy in Theorem 1, we show empirically that by using a proper guide-policy, the execute-policy can achieve good results.

---

> > ### Comment · Reviewer_PSxU · 2022-08-08
> > **Reviewer response**
> >
> > Thank you for answering my questions. I decide to increase my score to a 6.

---

> ### Author Response · Authors · 2022-08-07
> **Update**
>
> Dear reviewer,
>
> Please let us know if our response has addressed the issues raised in your review. We hope that our corrections, clarifications, and additional results address the concerns you've raised. We are happy to address any further concerns.

---

### Official Review · Reviewer_3s43 · 2022-07-12

**Rating:** 7
**Confidence:** 4
**Soundness:** 3 good
**Presentation:** 3 good
**Contribution:** 3 good

**Summary:**

This paper proposes a new algorithm for the offline RL problem (learning a high-performing policy from a fixed dataset instead of directly interacting with an environment). The main idea is to decompose the agent into a *guide-policy* —— which suggests the highest value next state to transition to given the current state —— and an *execute-policy*, which produces the action that is likely to transition the agent to a given state in one step. By using this decomposition, the paper claims to avoid the pitfalls of methods that use an action-value function, while retaining the stability of imitation learning methods.


**Questions:**

Please address the weaknesses listed above (and fix spelling).

**Limitations:**

$\alpha$ is a new "conservative-ness" hyperparameter similar to other offline RL algorithms, and it is not clear how to set it for a new use problem. If the authors have any algorithmic or practical suggestions for this, it would be great to have them included in the paper.


**Strengths And Weaknesses:**


Strengths:
- The proposed decomposition of the agent and training process into guide and execute policies is a new and interesting contribution.
- The main contributions of the paper are communicated clearly, with motivating examples and discussions to make the ideas understandable.
- The experiments are reasonably well designed and explore interesting properties of the proposed POR algorithm beyond benchmark scores. I particularly like Sec. 5.3 for giving readers an interesting insight into how POR can be useful in certain situations.

Weaknesses:
- The proposed execute-policy is presented as an instance of RL via supervised learning popular more recently (citation to UDRL [1] is missing here), and this framework is discussed multiple times, but what is actually learned is in fact known as an "inverse dynamics model", so it would be more relevant to cite references to prior work on those. [2] is a nice paper from 2016 that learned one using neural networks, but the concept is older and more general.
- The experiment in Sec. 5.3 is quite interesting, but it is unclear why the execute policy is not trained on $\mathcal{D}_o$. If I understand correctly, the execute policy is not a typical policy but an inverse dynamics model as mentioned above, and it can be trained on data from any policy.
- The paper contains a large number of spelling mistakes, which feels unprofessional. A spelling checker would easily find these.

[1] Srivastava, R.K., Shyam, P., Mutz, F., Jaśkowski, W. and Schmidhuber, J., 2019. Training agents using upside-down reinforcement learning. arXiv preprint arXiv:1912.02877.

[2] Christiano, P., Shah, Z., Mordatch, I., Schneider, J., Blackwell, T., Tobin, J., Abbeel, P. and Zaremba, W., 2016. Transfer from simulation to real world through learning deep inverse dynamics model. arXiv preprint arXiv:1610.03518.

---

> ### Author Response · Authors · 2022-08-02
> **Response to Reviewer 3s43**
>
> Thank you for your review and for providing relevant references that we had overlooked.
>
> >"so it would be more relevant to cite references to prior work on those"
>
> Thanks a lot for pointing out that! We have included a section to discuss more related work about the inverse dynamics model, see Appendix B for details.
>
> >"The experiment in Sec. 5.3 is quite interesting, but it is unclear why the execute policy is not trained on $\mathcal{D_{o}}$. If I understand correctly, the execute policy is not a typical policy but an inverse dynamics model as mentioned above, and it can be trained on data from any policy."
>
> While in principle the execute-policy can be trained on any data, we find in practice that there's a trade-off between the data coverage and data quality used to train the execute-policy. We conjecture the reason is that the neural networks we used are still not powerful enough to capture all different modes, those low-quality data may affect the network learning, especially when the size of high-quality data is small. Using more powerful network architecture/representation learning/more advanced supervised learning techniques might help.
>
> >"The paper contains a large number of spelling mistakes, which feels unprofessional"
>
> We are very sorry for this, we have checked and corrected all spelling mistakes in the revision.

---

### Meta-Review · Area_Chair_vTrD · 2022-08-20

**Recommendation:** Accept
**Confidence:** Certain

**Metareview:**

This paper proposes an interesting new idea that is well-motivated through illustrative examples and is thoroughly evaluated. There are some ways in which the paper could be improved, e.g. by including additional experiments (e.g. with high-dim observation spaces, transfer across action spaces, and discrete action spaces), but there don't appear to be any major weaknesses. The paper is clearly above the bar for acceptance at NeurIPS.

**Award:**

No

---

### Decision · Program_Chairs · 2022-09-14

Accept